# Ptbp1 is not required for retinal neurogenesis and cell fate specification

Haley Appel[1], Rogger P Carmen-Orozco[1], Clayton P Santiago[1,2], Thanh Hoang[3,4,5], Seth Blackshaw[6,7,8,9]*

[1]Solomon H. Snyder Department of Neuroscience, Johns Hopkins University School of Medicine, Baltimore, United States; [2]Department of Ophthalmology, University of Pittsburgh, Pittsburgh, United States; [3]Department of Ophthalmology and Visual Sciences, University of Michigan School of Medicine, Ann Arbor, United States; [4]Department of Cell and Developmental Biology, University of Michigan School of Medicine, Ann Arbor, United States; [5]Michigan Neuroscience Institute, University of Michigan School of Medicine, Ann Arbor, United States; [6]Department of Ophthalmology, Johns Hopkins University School of Medicine, Baltimore, United States; [7]Department of Neurology, Johns Hopkins University School of Medicine, Baltimore, United States; [8]Institute for Cell Engineering, Johns Hopkins University School of Medicine, Baltimore, United States; [9]Kavli Neuroscience Discovery Institute, Johns Hopkins University School of Medicine, Baltimore, United States

## eLife Assessment

This study used a conditional knockout mouse line to remove Ptbp1 in retinal progenitors and demonstrated that its deletion has no effect on retinal neurogenesis or cell fate specification, thereby challenging the prevailing view of Ptbp1 as a master regulator of neuronal fate. The data are **convincing**, supported by transcriptomic analysis, histology, and proliferation assays. This study is **important**, and the broader implications for other CNS regions warrant further investigation.

*For correspondence:
sblack@jhmi.edu

**Abstract** The RNA-binding protein Ptbp1 has been proposed as a master regulator of neuronal fate, repressing neurogenesis through its effects on alternative splicing and miRNA maturation. While prior studies using RNA interference suggested that Ptbp1 loss promotes neurogenesis, recent genetic studies have failed to replicate glia-to-neuron conversion following *Ptbp1* loss of function. To evaluate the role of Ptbp1 in developmental neurogenesis in vivo, we conditionally disrupted *Ptbp1* in mouse retinal progenitors. Ptbp1 was robustly expressed in both retinal progenitors and Müller glia but absent from postmitotic neurons, and efficient loss of function in mutant animals was confirmed using immunostaining for Ptbp1. Furthermore, bulk RNA-seq at E16 revealed accelerated expression of late-stage progenitor and photoreceptor-specific genes and altered splicing patterns in *Ptbp1* mutants, including increased inclusion of rod photoreceptor-specific exons. However, we observed no defects in retinal lamination, progenitor proliferation, or cell fate specification in mature retina. ScRNA-seq of mature mutant retinas revealed only modest transcriptional changes which partially recapitulate alterations seen following selective deletion of *Ptbp1* in mature glia. Our findings demonstrate that Ptbp1 is dispensable for retinal cell fate specification and suggest that its proposed role as a central repressor of neurogenesis should be reevaluated.

## Introduction

Control of neurogenesis in progenitor cells of the developing CNS involves an elaborate interplay between negative and positive regulators. Neurogenic bHLH factors such as Ascl1 and Neurog2 induce neural progenitors to undergo terminal neurogenic divisions, while inhibitory factors such as components of the Notch pathway and NFI family transcription factors promote proliferative divisions and gliogenesis (*Clark et al., 2019*; *Hufnagel et al., 2013*; *Keeley et al., 2023*; *Lyu et al., 2021*; *Yaron et al., 2006*). Studies in species such as zebrafish that undergo injury-induced neurogenesis by reprogramming of endogenous glial cells find that this process requires both upregulation of neurogenic factors and downregulation of inhibitory factors (*Briona et al., 2015*; *Fausett et al., 2008*; *Lyu et al., 2023*). This has recently inspired many studies aimed at inducing neurogenic competence in mammalian glia by selectively targeting these factors (*Hoang et al., 2020*; *Jorstad et al., 2017*; *Le et al., 2025*; *Le et al., 2024*; *Todd et al., 2021*; *Todd et al., 2022*).

One factor that has attracted considerable interest is the RNA-binding protein and splicing regulator Ptbp1, which is widely expressed in non-neuronal cells, including both neural progenitors and glia (*Boutz et al., 2007*; *Lilleväli et al., 2001*). It has been reported that Ptbp1 acts as a broad inhibitor of neuronal-specific splicing (*Boutz et al., 2007*; *Ling et al., 2016*), and inhibits maturation of miR-124a, which strongly promotes neurogenesis (*Makeyev et al., 2007*), while its paralog Ptbp2 promotes neuronal-specific splicing (*Licatalosi et al., 2012*; *Li et al., 2014*). It was also proposed that Ptbp1 acts as a master regulator of developmental neurogenesis in the CNS (*Shibasaki et al., 2013*). This then inspired studies which used *Ptbp1* knockdown to attempt to reprogram glia into functional neurons. While it was reported that *Ptbp1* knockdown in vivo induced robust generation of functional retinal ganglion cells from Müller glia (*Zhou et al., 2020*), as well as functional dopaminergic neurons from midbrain astrocytes (*Qian et al., 2020*), attempts to replicate these results by other groups were unsuccessful (*Chen et al., 2022*; *Hoang et al., 2023*; *Hoang et al., 2022*; *Wang et al., 2021*; *Xie et al., 2022*). Notably, genetic loss of function of *Ptbp1* coupled with ScRNA-seq analysis failed to observe either loss of expression of glial-specific genes or induction of neuron-specific genes in mutant glia (*Hoang et al., 2023*; *Hoang et al., 2022*).

This result raises questions about previous claims about the central role of Ptbp1 in controlling developmental neurogenesis. Previous functional studies of the role of Ptbp1 in neuronal development were primarily conducted in vitro using RNAi to interrogate Ptbp1 function (*Linares et al., 2015*; *Spellman et al., 2007*; *Zheng et al., 2012*). The few in vivo studies of Ptbp1 function in developing CNS have also almost exclusively used RNAi, and generally report modest and inconsistent phenotypes (*Zhang et al., 2016*). Conventional *Ptbp1* homozygous mutants are lethal, precluding direct analysis of its role in developmental neurogenesis (*Shibayama et al., 2009*). While one study reported accelerated cortical neurogenesis in *Ptbp1* heterozygotes (*Shibasaki et al., 2013*), these effects were also quite modest and difficult to interpret, given the broader role of Ptbp1 in early embryonic development.

The retina is a highly accessible and tractable model for the study of CNS neurogenesis and cell fate specification (*Ptito et al., 2021*), and conditional mutagenesis allows selective loss of function of any gene of interest in retinal progenitors without affecting viability (*Sauer, 1998*). We sought to use this approach to investigate the role of Ptbp1 in regulating neurogenesis and cell fate specification in developing retina. While we observe changes in RNA splicing and a moderate acceleration of retinal development as measured by bulk RNA-seq, we do not observe any changes in proliferation or neurogenesis during retinal development, or changes in the cell type composition of *Ptbp1*-deficient retinas. Furthermore, we observe only modest changes in gene expression in adolescent mutant retina using ScRNA-seq analysis. These results demonstrate that *Ptbp1* is dispensable for retinal cell fate specification.

## Results

### Ptbp1 mRNA is primarily expressed in retinal progenitors and Müller glia

To examine the expression patterns of *Ptbp1* and *Ptbp2* in developing and adult neuroretina, we analyzed ScRNA-seq datasets from mouse and human retina that had been previously generated by our lab (*Clark et al., 2019*; *Lu et al., 2020*). We observe high expression levels of *Ptbp1* mRNA in

primary retinal progenitors in both species and Müller glia in mouse retina, with weaker expression in neurogenic progenitors, and little expression detectable in neurons at any developmental age. In contrast, *Ptbp2* is expressed in both progenitors and neurons, although with somewhat higher expression in retinal ganglion and amacrine cells (*Figure 1—figure supplement 1a*). We observe that *Ptbp1* is expressed in primary progenitors at all developmental ages between E12 and P8 in mice and 9 and 20 gestational weeks in humans, although expression levels are moderately increased at later stages of neurogenesis (*Figure 1—figure supplement 1b*).

## *Chx10-Cre;Ptbp1*$^{lox/lox}$ mice show loss of Ptbp1 immunoreactivity in developing and mature retina

To selectively disrupt *Ptbp1* function in retinal progenitors, we generated *Chx10-Cre;Ptbp1*$^{lox/lox}$ mice. In these mice, Cre recombinase is expressed broadly in retinal progenitors from the beginning of neurogenesis onwards (*Rowan and Cepko, 2004*), allowing efficient deletion of the conditional mutant allele of *Ptbp1* (*Shibasaki et al., 2013*; *Figure 1a*). In some cases, the *Sun1-GFP* reporter line was used to visualize patterns of Cre activity (*Mo et al., 2015*). Mutant mice were born at expected Mendelian ratios and showed no gross defects in retinal morphology. To confirm efficient deletion of *Ptbp1* in retinal progenitors, we performed immunohistochemistry for Ptbp1 at E14, P1, and P30 in both control *Chx10-Cre;Ptbp1*$^{+/+}$ and homozygous mutant and *Chx10-Cre;Ptbp1*$^{lox/lox}$ and *Chx10-Cre;Ptbp1*$^{lox/lox}$;*Sun1-GFP mice* (*Figure 1b–d*). Perinuclear Sun1-GFP expression was observed in all retinal cells in *Chx10-Cre;Ptbp1*$^{lox/lox}$;*Sun1-GFP mice*, reflecting the broad retinal progenitor-specific activity of the *Chx10-Cre* transgene, while nuclear GFP expression was observed in retinal progenitors (*Mo et al., 2015*), reflecting the fact that a Cre-GFP fusion is used in the *Chx10-Cre* transgenic line (*Rowan and Cepko, 2004*). At all stages, we observed a loss of *Ptbp1* protein expression in retinal progenitors and Müller glia, with a more than 10-fold reduction in the number of immunoreactive Müller glia observed at P30 (*Figure 1e*). At both E14 and P1, we also observed a variable level of immunoreactivity in cells in the ganglion cell layer in control mice, which was also seen in *Chx10-Cre;Ptbp1*$^{lox/lox}$ mice, suggesting that this may represent cross-reactivity with an unknown epitope. This ganglion cell layer staining was not observed at P30, however. This demonstrates that *Ptbp1* is efficiently and selectively deleted in retinal progenitors from early stages of neurogenesis.

## Embryonic *Chx10-Cre;Ptbp1*$^{lox/lox}$ mice show shifts in temporal identity, increased expression of genes specific to photoreceptors, and altered patterns of RNA splicing

To further investigate the effects of *Ptbp1* deletion on embryonic progenitors, we conducted bulk RNA-seq on whole retina from E16 wildtype, *Chx10-Cre;Ptbp1*$^{lox/+}$, and *Chx10-Cre;Ptbp1*$^{lox/lox}$ retinas. A comparison of heterozygous *Chx10-Cre;Ptbp1*$^{lox/+}$ and homozygous *Chx10-Cre;Ptbp1*$^{lox/lox}$ mutants identified 2075 genes that were significantly differentially expressed (*Figure 2a*, *Supplementary file 1*). As expected, homozygotes showed a loss of *Ptbp1* expression along with a compensatory upregulation of *Ptbp2*, as has previously been observed following selective disruption of *Ptbp1* in mature Müller glia (*Hoang et al., 2022*; *Figure 2b*). In addition, we observed a modest increase in expression of multiple genes specific to both rods and cone photoreceptors, including *Opn1sw* and *Rcvrn*, as well as altered expression of multiple genes selectively expressed in either early or late-stage RPCs (*Clark et al., 2019*; *Lu et al., 2020*; *Lyu et al., 2021*). For instance, the early-stage RPC-specific genes *Sfrp2* and *Foxp1* were downregulated, while late-stage RPC-enriched genes such as *Sox8*, *Nfix*, and *Hes5* were upregulated (*Figure 2b*). Gene Ontology (GO) enrichment analysis likewise showed significant upregulation of genes involved in photoreceptor cell development and genes controlling synaptic function (*Figure 2c*). However, immunohistochemical analysis does not detect any clear changes in the expression of HuC/D+NeuN, Otx2 or Opn1sw in *Chx10-Cre;Ptbp1*$^{lox/lox}$ homozygous mutants relative to *Chx10-Cre;Ptbp1*$^{+/+}$ control at either E14 or P0, and the number of Otx2-positive cells at P0 was not altered (*Figure 2—figure supplement 1*). This implies that while at the transcriptional level retinal development in embryonic *Ptbp1* mutants appears to be moderately accelerated, based on higher expression of both markers of late-stage retinal progenitors and photoreceptors, this does not result in an increase in the number of photoreceptors at this stage.

In light of the established role of Ptbp1 in regulating RNA splicing in non-neuronal cells, we analyzed splicing differences in *Ptbp1*-deficient retinas using our dataset. We identified 864

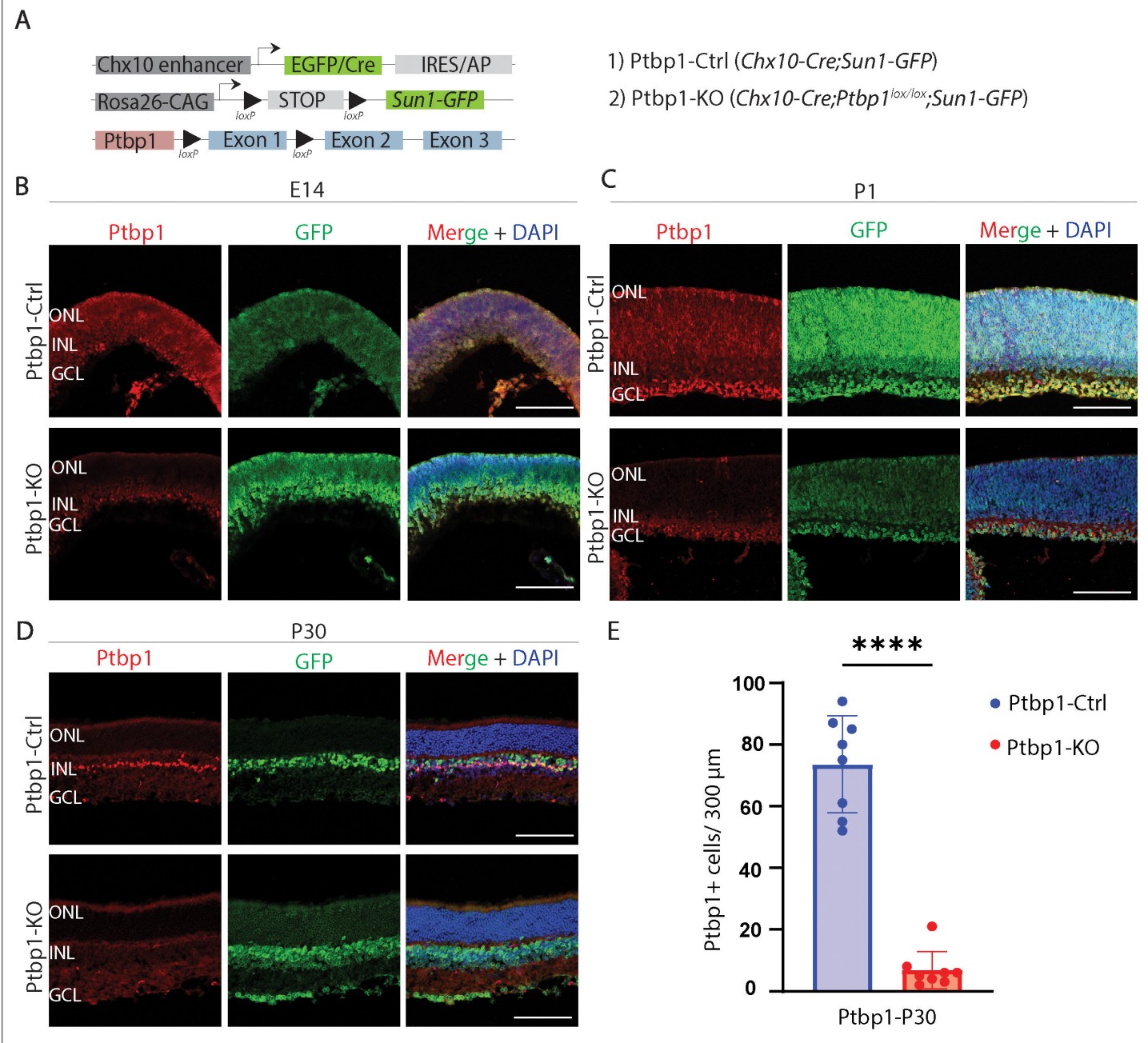

**Figure 1.** *Ptbp1* is successfully deleted in *Chx10-Cre;Ptbp1^{lox/lox}* mice in developing and mature retina. (**A**) A schematic diagram of the generation of specific deletion of *Ptbp1* in early RPCs using *Chx10-Cre*. (**B**) Representative immunostaining for *Ptbp1* expression in *Chx10-Cre;Ptbp1^{+/+}* (Ptbp1-Ctrl) and *Chx10-Cre;Ptbp1^{lox/lox}* (Ptbp1-KO) mice at E14.5, (**C**) *Chx10-Cre;Ptbp1^{+/+};Sun1-GFP* and *Chx10-Cre;Ptbp1^{lox/lox};Sun1-GFP* mice at P1, (**D**) and *Chx10-Cre;Ptbp1^{+/+}*, and *Chx10-Cre;Ptbp1^{lox/lox};Sun1-GFP* mice P30. (**E**) Quantification of Ptbp1-positive cells in Ptbp1-Ctrl and Ptbp1-KO retinas (n=8/genotype). Significance was determined via an unpaired *t*-test: **** *p*<0.0001. Each data point was calculated from an individual retina. ONL, outer nuclear layer; INL, inner nuclear layer; GCL, ganglion cell layer. Scale bar = 100 µm.

The online version of this article includes the following figure supplement(s) for figure 1:

**Figure supplement 1.** Cellular expression patterns of *Ptbp1* and *Ptbp2* during mouse and human retinal development.

differential percent-spliced-in (PSI) events, including changes in retained introns, alternative 5' or 3' splice site usage, mutually exclusive exons, and exon skipping, consistent with the known role of Ptbp1 as a splicing repressor (*Figure 2d–e*, *Supplementary file 1*; *Vuong et al., 2016*; *Yap et al., 2012*). To assess whether these splicing changes reflect neuronal or rod photoreceptor maturation,

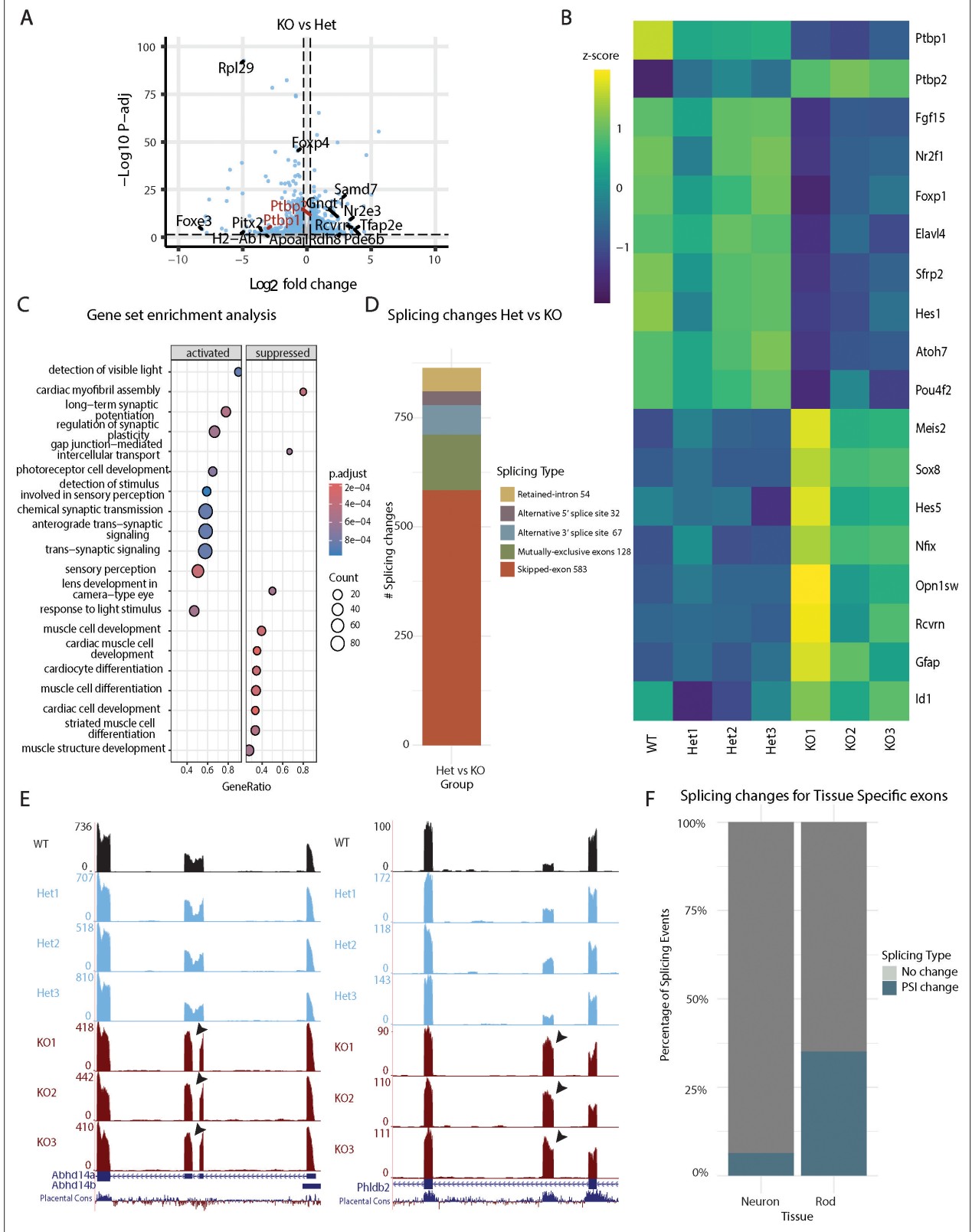

**Figure 2.** Bulk RNA-seq analysis reveals accelerated differentiation and altered RNA splicing in *Chx10-Cre;Ptbp1^lox/lox* mice at E16.5. (**A**) Volcano plot showing 2075 differentially expressed genes between heterozygous *Chx10-Cre;Ptbp1^lox/+* and homozygous *Chx10-Cre;Ptbp1^lox/lox* mutants as measured by bulk RNA-seq. (**B**) Heatmap of select differentially expressed genes in wildtype *Chx10-Cre;Ptbp1^+/+*, heterozygous *Chx10-Cre;Ptbp1^lox/+*, and homozygous *Chx10-Cre;Ptbp1^lox/lox* samples. (**C**) Gene set enrichment analysis showing activated and repressed gene ontology terms in the homozygous

*Figure 2 continued on next page*

*Figure 2 continued*

*Ptbp1* mutant samples. (**D**) Barplot showing 864 splicing changes in homozygous *Ptbp1* mutant samples by splicing class, including retained introns (54 events), alternative 5' splice site (32 events), alternative 3' splice site (67 events), mutually exclusive exons (128 events) and skipped exons (583 events). (**E**) Track plots showing examples of alternative splicing of exons (arrows) in the *Ptbp1* mutant samples. (**F**) Bar plot showing that **7%** of neuron-enriched and **35%** of rod-specific splicing events overlap with splicing changes observed in *Ptbp1* mutants.

The online version of this article includes the following figure supplement(s) for figure 2:

**Figure supplement 1.** Immunostaining does not detect changes in cell composition in E14 or P0 *Ptbp1*-deficient retina.

**Figure supplement 2.** Bulk RNA-seq analysis reveals altered gene expression and splicing patterns in E16 *Chx10-Cre;Ptbp1^{lox/lox}* mice retina.

we compared our differential splicing events to those observed in adult neurons and rods (***Ling et al., 2020***). We found that 8 splicing events (7%) known to be enriched in central nervous system neurons and 19 events (35%) specific to rods overlapped with the splicing changes observed in *Ptbp1*-deficient retinas (***Figure 2f, Supplementary file 2***).

Gene expression and splicing changes were compared across several reference tissues: heart tissue and Thy1-positive neurons, mature hair cells, microglia, and astrocytes (***Figure 2—figure supplement 2a and b***). A heatmap of differentially expressed genes showed that while *Ptbp1*-deficient retinas diverged from WT retinas, their expression profiles did not resemble those of fully differentiated cell types like rods, astrocytes, or adult WT retina (***Figure 2—figure supplement 2c***). Consistently, Pearson correlation analysis revealed that *Ptbp1*-deficient and WT retinas were more like each other than to fully differentiated neuronal or glial populations (***Figure 2—figure supplement 2d***). Splicing profile analysis further revealed that while there was a high correlation of PSI between *Ptbp1*-deficient and WT retinas, *Ptbp1*-deficient retinas more closely resembled Thy1-positive neurons, whereas WT retinas aligned more strongly with mature cells such as astrocytes, microglia, and auditory hair cells (***Figure 2—figure supplement 2e–f***). Together, these results suggest that although *Ptbp1* loss of function induces hundreds of alternative splicing events, the magnitude of PSI changes in the KO retinas remains considerably lower than that seen in fully differentiated cell types (***Supplementary file 1***). Thus, while a subset of splicing events overlaps with those characteristic of mature neurons or rods, the overall splicing and expression profiles of KO retinas are more like those of developing retinal tissue rather than terminally differentiated neuronal or glial populations.

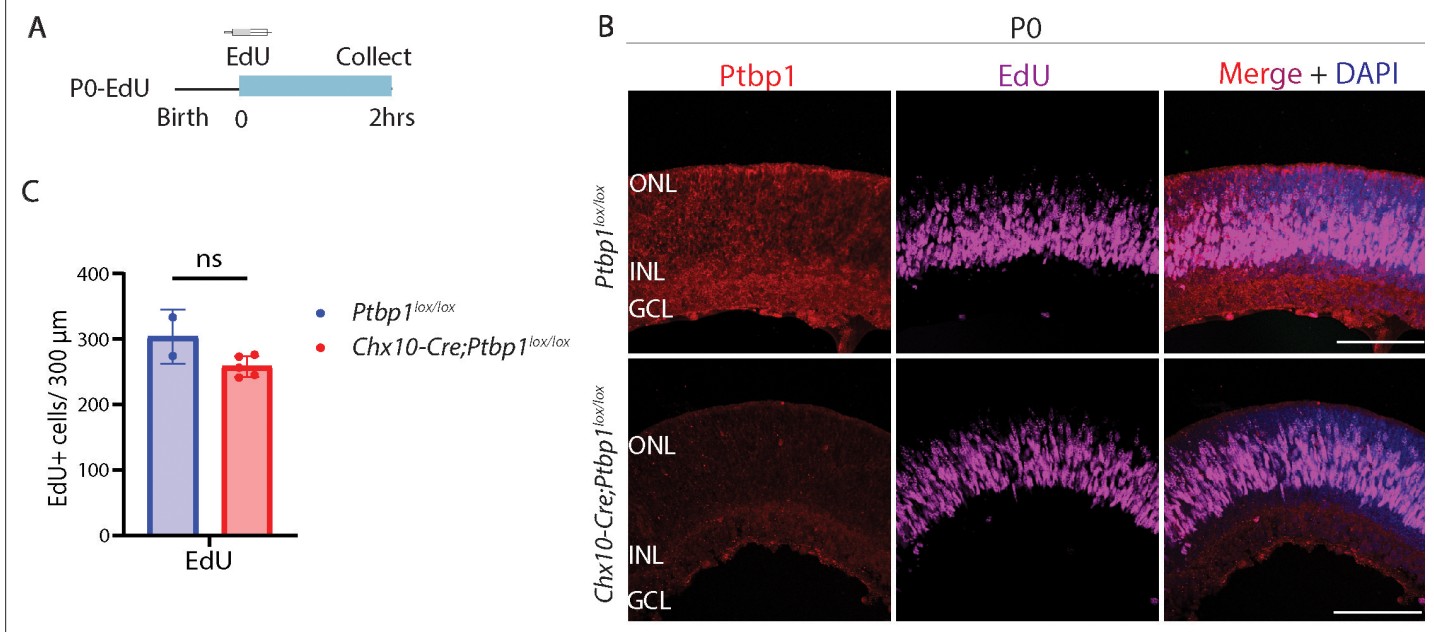

**Figure 3.** *Ptbp1* deletion does not alter retinal progenitor proliferation. (**A**) A schematic diagram of the experimental timeline for labeling dividing cells using EdU. (**B**) Representative immunostaining for Ptbp1 expression and EdU labeling in Ptbp1-Ctrl and Ptbp1-KO mice at P0. (**C**) Quantification of EdU-positive cells in Ptbp1-Ctrl and Ptbp1-KO retinas (n≥2/genotype). Significance was determined via the Mann-Whitney test: ns = p>0.05. Each data point was calculated from an individual retina. ONL, outer nuclear layer; INL, inner nuclear layer; GCL, ganglion cell layer. Scale bar = 100 μm.

## *Chx10-Cre;Ptbp1^{lox/lox}* mice show no changes in retinal progenitor proliferation or neurogenesis

To determine whether retinal progenitor-specific loss of function of *Ptbp1* resulted in a depletion of retinal progenitors, we conducted EdU labeling to quantify the number of cells in S-phase in P0 retina (*Figure 3a*). We did not observe any significant differences in the number of EdU-positive cells between Cre negative *Ptbp1^{lox/lox}* controls and *Chx10-Cre;Ptbp1^{lox/lox}* mice (*Figure 3b and c*). To determine whether loss of function of *Ptbp1* influenced cell type specification in developing retina, we conducted immunostaining for a range of cell type-specific markers in *Chx10-Cre* control and *Chx10-Cre;Ptbp1^{lox/lox}* mice (*Figure 4*). Although Ptbp1 immunoreactivity was lost in homozygous mutant retinas (*Figure 4a*), we did not observe any significant changes in the numbers or distribution of Sox9-positive Müller glia (*Figure 4b and f*), HuC/D-positive amacrine and ganglion cells (*Figure 4c and f*), Otx2-positive bipolar cells and photoreceptors (*Figure 4c and f*), Rbpms-positive retinal ganglion cells (*Figure 4d and f*), or cone arrestin-positive cone photoreceptors (*Figure 4e and f*). Taken together, these findings show that *Ptbp1* loss of function induces neither precocious cell cycle exit nor affects normal levels and patterns of developmental neurogenesis and cell fate specification.

## *Chx10-Cre;Ptbp1^{lox/lox}* mice show only modest changes in gene expression in the mature retina

Despite the lack of any obvious defects in proliferation or cell fate specification in *Ptbp1*-deficient retinas, it nonetheless remained possible that this mutant might generate more subtle defects that are detectable using ScRNA-seq analysis of whole retina. We, therefore, generated ScRNA-seq libraries from both P30 *Chx10-Cre;Ptbp1^{+/+}* and *Chx10-Cre;Ptbp1^{lox/lox}* mice. No changes in the identity or composition of any retinal cell type were observed (*Figure 5a–c*). The overall identities and proportions of major retinal cell types appeared broadly similar between control and knockout samples (*Figure 5a–c*). A modest number of genes in rod and cone photoreceptors, as well as bipolar neurons, showed statistically significant changes in gene expression (*Figure 4—figure supplement 1*, *Supplementary file 3*). A subset of phototransduction (*Pde6g*, *Guca1a*, *Pdc*, *Rcvrn*), photoreceptor-specific structural components (*Rom1*), and rod-specific transcription factors *Nrl* and *Nr2e3* all showed modest but significant increases in expression in rods, while transcription factors such as *Otx2* and *Prdm1*, which are more prominently expressed in immature photoreceptors, showed significantly reduced expression. Cone photoreceptors showed increased expression of some phototransduction components (*Pde6c*, *Rcvrn*, *Guca1a*) and decreased expression of others (*Arr3*).

The largest number of differentially expressed genes were observed in Müller glia (*Figure 5d*). Several of the most strongly upregulated genes – notably including *Ptbp2*, *Fxyd6*, *Mt1*, and *Mt3* – were previously found to be upregulated by selective loss of function of *Ptbp1* in mature Müller glia in *Glast-CreER;Ptbp1^{lox/lox}* mice, with nearly a third of all significantly downregulated genes shared between both mutants (*Hoang et al., 2022*; *Figure 5e*, *Supplementary file 3*). However, no significant changes in canonical markers of Müller glia such as *Sox9*, *Apoe*, *Glul*, or *Rlbp1* were observed (*Figure 5f*).

## Discussion

Previous studies have reported that Ptbp1 acts in neural progenitors in the CNS as a master repressor of neuronal fate (*Hu et al., 2018*; *Shibasaki et al., 2013*). If this were the case, we would expect that *Ptbp1* loss of function in early development would rapidly lead to precocious cell cycle exit and reduction in the number of progenitor cells. As a result, we would expect to observe an overrepresentation of early-born cell types such as retinal ganglion cells and cone photoreceptors at the expense of late-born cell types such as bipolar interneurons and, in particular, Müller glia. However, we observe that while Ptbp1 is robustly expressed in both retinal progenitors and mature glia, as in other CNS regions, loss of *Ptbp1* function in early-stage retinal progenitors does not induce any significant developmental defects. Despite efficient and widespread loss of Ptbp1 expression from E14 onwards, we do not observe any changes in progenitor proliferation, retinal cell type composition, or expression of canonical Müller glia marker genes in mature retina. The changes in gene expression that are observed in adolescent *Ptbp1*-deficient retinas are quite modest, although increased levels of several photoreceptor-specific genes are observed. This is consistent with previous studies showing

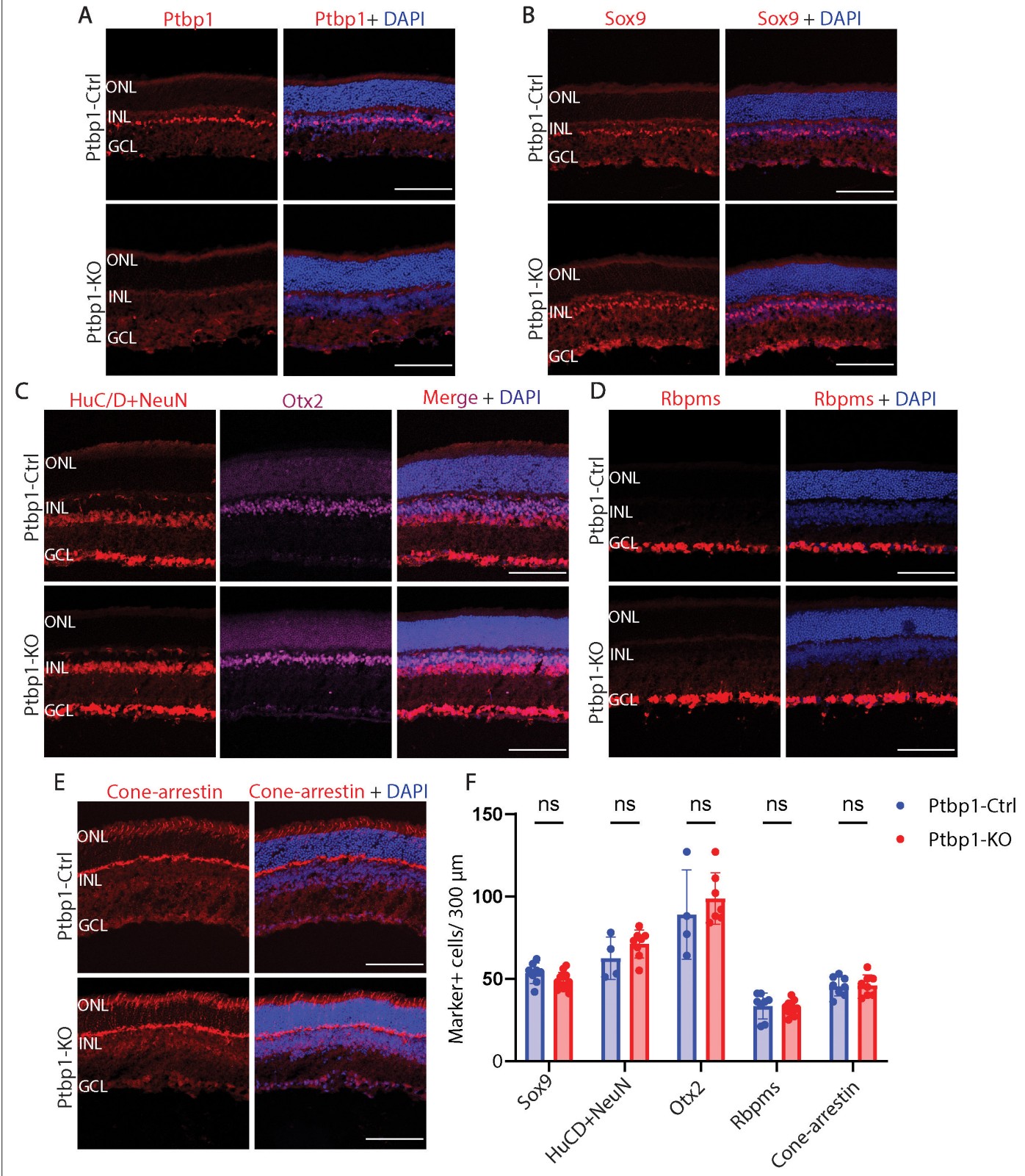

**Figure 4.** *Ptbp1* deletion does not alter retinal neurogenesis. (**A**) Representative immunostaining for Ptbp1, (**B**) Sox9, (**C**) HuC/D+NeuN, Otx2, (**D**) Rbpms, and (**E**) Cone-arrestin in Ptbp1-Ctrl and Ptbp1-KO mice at P30. (**F**) Quantification of Sox9, HuC/D/NeuN, Otx2, Rbpms, and Cone-arrestin-positive cells in Ptbp1-Ctrl and Ptbp1-KO retinas (n≥4/genotype). Significance was determined via multiple unpaired *t* tests: ns = p>0.05. Each data point was calculated from an individual retina. ONL, outer nuclear layer; INL, inner nuclear layer; GCL, ganglion cell layer. Scale bar = 100 μm.

*Figure 4 continued on next page*

*Figure 4 continued*

The online version of this article includes the following figure supplement(s) for figure 4:

**Figure supplement 1.** Single-cell RNA-sequencing (ScRNA-seq) analysis identifies differentially expressed genes in major adolescent retinal cell types following developmental loss of function of *Ptbp1*.

that *Ptbp1* knockdown enhances photoreceptor-specific splicing of a subset of alternatively spliced genes (*Ling et al., 2020*), and implies that this may indirectly increase expression of a subset of rod-specific genes.

At E16, however, homozygous *Ptbp1* mutants do show changes in gene expression consistent with previously reported functions. These retinas show increased expression of *Ptbp2*, the neuronally enriched paralog of *Ptbp1*, along with increased expression of genes specific to late-stage progenitors and photoreceptor precursors. Furthermore, *Ptbp1* mutants exhibit altered RNA splicing and inclusion of a subset of exons specific to rod photoreceptors. While this is apparently consistent with reports that developmental loss of *Ptbp1* function promotes neurogenesis by inducing a neuronal-like splicing pattern in neural progenitors (*Boutz et al., 2007*; *Hu et al., 2018*; *Shibasaki et al., 2013*; *Vuong et al., 2016*), this does not result in changes in either progenitor proliferation or any change in retinal cell type composition in either the developing or adolescent retina. This argues against the idea that *Ptbp1* loss simply accelerates the timing of neurogenesis, as there is no evidence for increased photoreceptor production or altered proliferation in embryonic or neonatal retina. An alternative interpretation would be that *Ptbp1* loss of function leads to the precocious formation of ectopic neurons which rapidly undergo cell death. Further studies will be needed to resolve this question.

Following closely behind studies that fail to show any role for Ptbp1 in regulating glia-to-neuron reprogramming in adult CNS (*Chen et al., 2022*; *Hoang et al., 2023*; *Hoang et al., 2022*; *Wang et al., 2021*), these findings call into question previous models that place Ptbp1 in a central position in controlling developmental neurogenesis. The reason for this discrepancy is not entirely clear. Previous studies using knockdown may have been complicated by off-target effects (*Jackson et al., 2003*), and conditions for in vitro analysis may not have accurately replicated conditions in the native CNS. It is also possible that compensatory mechanisms differ between knockdown and knockout approaches. Notably, a recent study (*Konar et al., 2025*) reported that *Ptbp1* knockdown promotes Müller glia proliferation in zebrafish, suggesting that the effects of acute reduction of *Ptbp1* mRNA may not fully mirror those of complete loss of function. Alternatively, upregulation of the widely expressed paralog *Ptbp2* may have compensated for developmental defects resulting from *Ptbp1* loss of function. Generation of conditional double mutant mice will be necessary to definitively exclude this possibility. Additionally, the use of *Chx10-Cre* in this study does not exclude a potential role for *Ptbp1* during early stages of retinal development prior to E11, when the Cre is first expressed. Regardless, this study demonstrates that *Ptbp1* is dispensable for the process of retinal cell fate specification, and we anticipate that this will likely hold for other CNS regions.

## Methods
### Mice
Mice were raised and housed in a climate-controlled pathogen-free facility on a 14/10 hr light/dark cycle. Mice used in this study were *Chx10-Cre* (JAX #005105) *Ptbp1^lox/lox^* mice carrying loxP sites that flank the promoter and 1st exon of *Ptbp1* were generated as described previously (*Shibayama et al., 2009*). *Chx10-Cre;Ptbp1^lox/lox^* mice were generated by crossing *Chx10-Cre* with conditional *Ptbp1^lox/lox^* mice. *Chx10-Cre;Ptbp1^lox/lox^;Sun1-GFP* mice were also examined in some cases to visualize Cre activity in postmitotic neurons (*Mo et al., 2015*). Maintenance and experimental procedures performed on mice were in accordance with the protocol approved by the Institutional Animal Care and Use Committee (IACUC) at the Johns Hopkins School of Medicine under protocol number MO22M22.

### EdU treatment
*Chx10-Cre* control and *Chx10-Cre;Ptbp1^lox/lox^* neonatal mice were administered EdU (Abcam, #ab146186) via subcutaneous injection at P0 [10–20 µl of 5 mg/ml in PBS]. Retinas were collected 2 hr post-EdU injection for analysis.

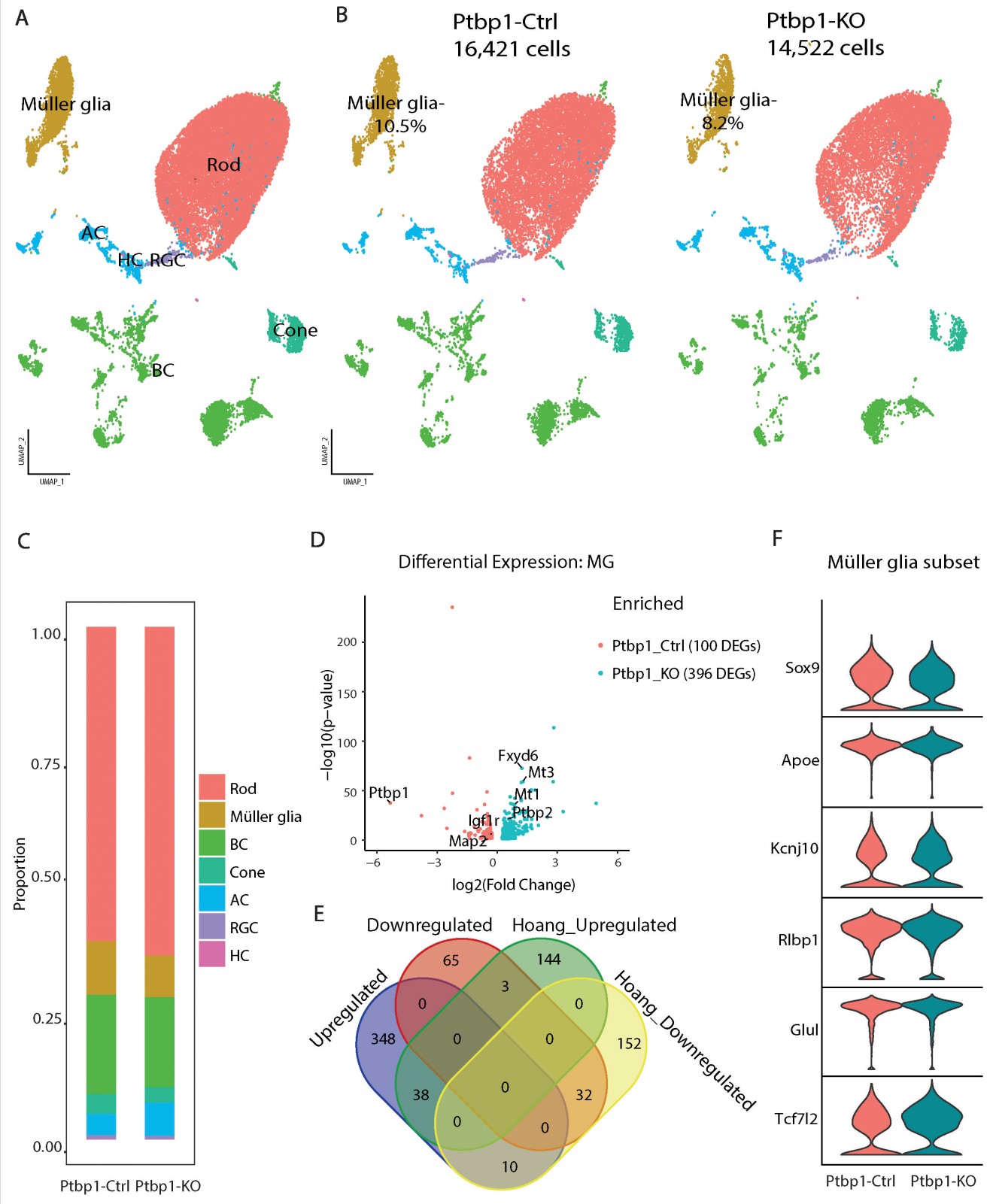

**Figure 5.** Single-cell RNA-sequencing (ScRNA-seq) analysis reveals modest transcriptional changes in Müller glial gene expression in *Chx10-Cre;Ptbp1*^lox/lox^ mice. (**A**) Uniform manifold approximation and projection (UMAP) plots showing cell types captured in the *Chx10-Cre;Ptbp1*^+/+^ (Ptbp1-Ctrl) and *Chx10-Cre;Ptbp1*^lox/lox^ (Ptbp1-KO) merged dataset. (**B**) UMAP plots showing cell types separated by genotype with the percentage of Müller glia in each group. (**C**) Barplot showing cell proportions for each cell type in both genotypes. (**D**) Volcano plot showing differentially expressed genes in

*Figure 5 continued*

Müller glia between Ptbp1-Ctrl and Ptbp1-KO samples. (**E**) Venn diagram comparison of upregulated and downregulated differentially expressed genes for Müller glia between this dataset and previously published data analyzing Müller glia-specific loss of function of *Ptbp1* (*Hoang et al., 2022*). (**F**) Violin plot showing no significant changes in the expression of canonical Müller glia markers in Ptbp1-KO sample compared to Ptbp1-Ctrl.

## Immunohistochemistry and imaging

Collection and immunohistochemical analysis of retinas were performed as described previously (*Hoang et al., 2020*). Briefly, mouse eye globes were fixed in 4% paraformaldehyde (Electron Microscopy Sciences, #15710) for 1 hr at 4 °C. Retinas were dissected in 1 x PBS and incubated in 30% sucrose overnight at 4 °C.

Retinas were then embedded in OCT (VWR, #95057–838), cryosectioned at 16 µm thickness, and stored at −20 °C. Sections were dried for 30 min in a 37 °C incubator and washed 3×5 min with 0.1% TritonX-100 in PBS (PBST). EdU labeling was performed by using the Click-iT EdU kit (Thermo Fisher, #C10340) following the manufacturer's instructions. Sections were then incubated in 10% Horse Serum (Thermo Fisher, #26050070), 0.4% Triton X-100 in 1 x PBS (blocking buffer) for 2 hr at room temperature (RT) and then incubated with primary antibodies in the blocking buffer overnight at 4 °C. Primary antibodies used in this study were: rabbit anti-Ptbp1 (1:200, Invitrogen, #PA3-81297), rabbit anti-Ptbp1 (1:200, ABclonal, #A6107), rabbit anti-RBPMS (1:200, Proteintech, #151871-AP), rabbit anti-cone arrestin (1:200, Millipore Sigma, #AB15282), goat anti-OTX2 (1:200, R&D Systems, #AF1979), mouse anti-HuC/D (1:200, Thermo Fisher Scientific, #A21271), mouse anti-NeuN (1:200, SigmaAldrich, #MAB377), rabbit anti-SOX9 (1:200, SigmaAldrich, #AB5535), and chicken anti-GFP (1:400, Thermo Fisher Scientific, #A10262). An equal mixture of anti-HuC/D and NeuN antibodies was used to detect amacrine and ganglion cells.

Sections were washed 3×5 min with PBST to remove excess primary antibodies and were incubated in secondary antibodies in blocking buffer for 2 hr at RT. Sections were then counterstained with DAPI in PBST, washed 3×5 min in PBST and mounted with ProLong Gold Antifade Mountant (Invitrogen, #P36935) under coverslips (VWR, #48404–453), air-dried, and stored at 4 °C. Fluorescent images were captured using a Zeiss LSM 700 confocal microscope.

## Cell quantification and statistical analysis

Ptbp1-positive, Sox9-positive, HuC/D/NeuN-positive, Otx2-positive, Rbpms-positive, and ArrestinC-positive cells were counted in total from 300 µM sections per retina. For cell proliferation quantification, EdU-positive cells were counted from 300 µM sections per retina. Each data point in the bar graphs was calculated from an individual retina. All cell quantification data were graphed and analyzed using GraphPad Prism 10. Multiple unpaired *t* tests or Mann-Whitney tests were used for analysis between samples of multiple groups. In all tests, values of $p<0.05$ were considered to indicate significance.

## Retinal cell dissociation

Briefly, one female mouse per genotype was euthanized by $CO_2$. Retinas were dissected in fresh ice-cold PBS and retinal cells were dissociated using Papain Dissociation System (LK003150, Worthington). Dissociated cells were resuspended in ice-cold HBAG buffer containing 9.8 mL HibernateA (BrainBits, #HALF500), 200 mL B27 (ThermoFisher, #17504044), 20 mL GlutaMAX (ThermoFisher, #35050061), and 0.5 U/mL RNAse inhibitor. Cells were filtered through a 50 mm filter. Cell count and viability were determined by using 0.4% Trypan blue.

## Bulk RNA-sequencing library preparation

Retinas were dissected from E16 mice embryos and stored at −80° C. Frozen retinal tissue was pooled into seven samples by genotype: wildtype (WT (six pooled retinas)); heterozygous *Ptbp1* mutant *Chx10-Cre;Ptbp1*<sup>lox/+</sup> (Het1 (five pooled retinas), Het2 (five pooled retinas), Het3 (six pooled retinas)); and homozygous *Ptbp1* mutant (KO1 (four pooled retinas), KO2 (four pooled retinas), and KO3 (six pooled retinas)). RNA was extracted using the RNeasy Micro Plus kit (Qiagen). Illumina Stranded mRNA Prep Ligation kit was used for bulk RNA library preparation. Libraries were sequenced on the Illumina NovaSeq X.

## Single-cell RNA-sequencing (scRNA-seq) library preparation

ScRNA-seq were prepared on dissociated retinal cells using the 10 X Genomics Chromium Single Cell 3' Reagents Kit v3.1 (10X Genomics, Pleasanton, CA). Libraries were constructed following the manufacturer's instructions and were sequenced using the Illumina NovaSeq 6000. The sequencing data were aligned to the mm10 reference genome and a cell-by-gene matrix was generated using the Cell Ranger 7.0.1 pipeline (10 X Genomics).

## ScRNA-seq data analysis

ScRNA-seq of control *Chx10-Cre;Ptbp1*$^{+/+}$ and homozygous mutant *Chx10-Cre;Ptbp1*$^{lox/lox}$ mice were analyzed using the Seurat package (*Hao et al., 2021*). Briefly, cells were filtered to retain those with more than 200 and fewer than 5000 detected genes, more than 700 and fewer than 10,000 total transcripts, and less than 20% mitochondrial gene content. The data were normalized using the default log-normalization parameters, the top 4000 highly variable features were identified, and the data were scaled while regressing out mitochondrial gene content. Principal component analysis (PCA) was then performed on the variable genes followed by Louvain clustering and UMAP visualization using the top 12 principal components. Expression of key marker genes within each cell cluster were used to confirm the appropriate assignment of cell types. Differentially expressed genes between groups were determined using the Wilcoxon rank sum test. The clusterProfiler R package was used to perform gene ontology enrichment analysis of biological processes found in the differentially expressed gene lists (*Yu et al., 2012*).

## Bulk RNA-seq analysis

Reads were processed using the rMATS-turbo pipeline (*Wang et al., 2024*), which performs alignment with STAR and custom splice junction counts. Alignments were generated using the GRCm38 mouse genome (Release M10) with GENCODE annotation. Gene-level counts were then extracted from the resulting BAM files using featureCounts, and differential gene expression was analyzed with the DESeq2 package in R (*Liao et al., 2014*; *Love et al., 2014*). Comparisons between *Ptbp1* heterozygous (Het) and knockout (KO) samples identified differentially expressed genes based on an absolute log2 fold change greater than 0.263 (representing at least a 20% change in expression) and an adjusted p-value below 0.05. We incorporated publicly available bulk RNA-seq datasets for comparative analysis, including embryonic day 16 retina (E16: GSM2720095, GSM2720096), postnatal day 28 retina (P28: GSM2720111, GSM2720112), Thy1-positive neurons (GSM2392791), astrocytes (GSM1269903, GSM1269904), microglia (GSM1269913, GSM1269914), hair cells (GSM1602228, GSM1602229), and heart tissue (GSM1223635) (*Brooks et al., 2019*; *Cai et al., 2015*; *Giudice et al., 2014*; *Yang et al., 2017*; *Zhang et al., 2014*). Hair cells and heart tissue samples were chosen as examples of high and low Ptbp1 expression, respectively, based on data from ASCOT (*Ling et al., 2020*). We generated a Pearson correlation matrix to assess overall sample similarity. Splicing differences were evaluated using filtered rMATS output, applying the following thresholds: read coverage ≥15, minimum PSI = 0.05, maximum PSI = 0.95, significant FDR (sigFDR)≤0.05, |ΔPSI|≥0.10, background FDR (bgFDR)≤0.5, and background within-group ΔPSI ≤0.2. For the E16 and P28 retina datasets, we used more relaxed filters (read coverage ≥0, PSI between 0.01 and 0.99) to ensure inclusion of lower-abundance events. Splicing events specific to rods and various neuronal subtypes were queried from the ASCOT supplementary files and visually inspected using the UCSC Genome Browser to assess their presence in bulk RNA-seq datasets. This analysis followed the approach described in previous studies (*Carmen-Orozco et al., 2024*; *Ling et al., 2020*).

## Acknowledgements

This work was supported by NIH grant R01EY036173 to SB.

## Additional information

### Competing interests

Seth Blackshaw: Cofounder and shareholder of CDI Labs, LLC, and receives research support from Genentech. The other authors declare that no competing interests exist.

## Funding

| Funder | Grant reference number | Author |
|---|---|---|
| National Eye Institute | R01EY036173 | Seth Blackshaw |

The funders had no role in study design, data collection and interpretation, or the decision to submit the work for publication.

## Author contributions

Haley Appel, Conceptualization, Data curation, Formal analysis, Validation, Investigation, Visualization, Methodology, Writing – original draft, Writing – review and editing; Rogger P Carmen-Orozco, Data curation, Formal analysis, Investigation, Methodology, Writing – original draft, Writing – review and editing; Clayton P Santiago, Data curation, Formal analysis, Supervision, Methodology, Writing – original draft, Writing – review and editing; Thanh Hoang, Conceptualization, Supervision, Investigation, Methodology, Writing – original draft, Writing – review and editing; Seth Blackshaw, Conceptualization, Supervision, Funding acquisition, Writing – original draft, Project administration, Writing – review and editing

## Author ORCIDs

Rogger P Carmen-Orozco (ID) https://orcid.org/0000-0002-2460-9270
Clayton P Santiago (ID) https://orcid.org/0000-0001-7191-668X
Thanh Hoang (ID) https://orcid.org/0000-0002-7666-3842
Seth Blackshaw (ID) https://orcid.org/0000-0002-1338-8476

## Ethics

Maintenance and experimental procedures performed on mice were in accordance ARRIVE guidelines for animal research, with the protocol approved by the Institutional Animal Care and Use Committee (IACUC) at the Johns Hopkins School of Medicine under protocol number MO22M22.

Reviewer #1 (Public review): https://doi.org/10.7554/eLife.108331.3.sa1
Reviewer #2 (Public review): https://doi.org/10.7554/eLife.108331.3.sa2
Author response https://doi.org/10.7554/eLife.108331.3.sa3

---

# Additional files

## Supplementary files

Supplementary file 1. Bulk RNA-seq analysis. Table of 2075 differentially expressed genes between heterozygous *Chx10-Cre;Ptbp1*<sup>lox/l+</sup> and homozygous *Chx10-Cre;Ptbp1*<sup>lox/lox</sup> mutants identified by bulk RNA-seq at E16.5, table of 864 differential splicing changes, and table of adult neuronal and rod-enriched splicing events that overlap with the differential splicing events in the bulk RNA-seq dataset.

Supplementary file 2. Bulk RNA-seq analysis of rod and neuron-specific exons. Genome-browser view showing exon-level read coverage for 18 Rod-specific (A) and eight Neuron-specific (B) exons across WT, heterozygous *Chx10-Cre;Ptbp1*<sup>lox/l+</sup>, and homozygous *Chx10-Cre;Ptbp1*<sup>lox/lox</sup> mutant retinas and additional tissue, including brain, rod photoreceptors (P2, P6, P14, P28), and various neuronal and glial cell types. Tracks indicate RNA-seq read density aligned to the mm10 genome, highlighting exon-specific differences in expression and potential alternative splicing events across genotypes.

Supplementary file 3. Single-cell RNA-sequencing (ScRNA-seq) analysis. Proportions of cell types in the Ptbp1-Ctrl and Ptbp1-KO ScRNA-seq datasets. Table of differentially expressed genes between *Chx10-Cre;Ptbp1*<sup>+/+</sup> control and homozygous *Chx10-Cre;Ptbp1*<sup>lox/lox</sup> mutants identified by ScRNA-seq in adolescent retina for Müller glia (496), bipolar cells (433), amacrine cells (138), rod photoreceptor (345), and cone photoreceptor cells (74) with adjusted p-value of 0.05. Positive avg_log2FC scores represent upregulated genes within the Ptbp1-KO sample.

MDAR checklist

## Data availability

All raw RNA-Seq and scRNA-Seq data are publicly available through the Gene Expression Omnibus under accession numbers GSE300588 and GSE300607. Analysis scripts used in this study are accessible at: https://github.com/csanti88/ptbp1_2025 (copy archived at *Santiago, 2025*). Bulk RNA-seq and other tissue track plots visualizations are available at: https://genome.ucsc.edu/s/roggercarmen/Ptbp1KO_Retina.

The following datasets were generated:

| Author(s) | Year | Dataset title | Dataset URL | Database and Identifier |
|---|---|---|---|---|
| Appel A, Carmen-Orozco RP, Santiago CP, Hoang T, Blackshaw S | 2025 | Ptbp1 is not required for retinal neurogenesis and cell fate specification | https://www.ncbi.nlm.nih.gov/geo/query/acc.cgi?acc=GSE300588 | NCBI Gene Expression Omnibus, GSE300588 |
| Appel A, Carmen-Orozco RP, Santiago CP, Hoang T, Blackshaw S | 2025 | Ptbp1 is not required for retinal neurogenesis and cell fate specification | https://www.ncbi.nlm.nih.gov/geo/query/acc.cgi?acc=GSE300607 | NCBI Gene Expression Omnibus, GSE300607 |

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
