## [Editor Report · eLife Assessment]

This study used a conditional knockout mouse line to remove Ptbp1 in retinal progenitors and demonstrated that its deletion has no effect on retinal neurogenesis or cell fate specification, thereby challenging the prevailing view of Ptbp1 as a master regulator of neuronal fate. The data are **convincing**, supported by transcriptomic analysis, histology, and proliferation assays. This study is **important**, and the broader implications for other CNS regions warrant further investigation.

---

## [Referee Report · Reviewer #1 (Public review)]

Summary:

The researchers sought to determine whether Ptbp1, an RNA-binding protein formerly thought to be a master regulator of neuronal differentiation, is required for retinal neurogenesis and cell fate specification. They used a conditional knockout mouse line to remove Ptbp1 in retinal progenitors and analyzed the results using bulk RNA-seq, single-cell RNA-seq, immunohistochemistry, and EdU labeling. Their findings show that Ptbp1 deletion has no effect on retinal development, since no defects were found in retinal lamination, progenitor proliferation, or cell type composition. Although bulk RNA-seq indicated changes in RNA splicing and increased expression of late-stage progenitor and photoreceptor genes in the mutants, and single-cell RNA-seq detected relatively minor transcriptional shifts in Müller glia, the overall phenotypic impact was low. As a result, the authors conclude that Ptbp1 is not required for retinal neurogenesis and development, thus contradicting prior statements about its important role as a master regulator of neurogenesis. They argue for a reassessment of this stated role. While the findings are strong in the setting of the retina, the larger implications for other areas of the CNS require more investigation. Furthermore, questions about potential reimbursement from Ptbp2 warrant further research.

Strengths:

This study calls into doubt the commonly held belief that Ptbp1 is a critical regulator of neurogenesis in the CNS, particularly in retinal development. The adoption of a conditional knockout mouse model provides a reliable way for eliminating Ptbp1 in retinal progenitors while avoiding the off-target effects often reported in RNAi experiments. The combination of bulk RNA-seq, scRNA-seq, and immunohistochemistry enables a thorough examination of molecular and cellular alterations at both embryonic and postnatal stages, which strengthens the study's findings. Furthermore, using publicly available RNA-Seq datasets for comparison improves the investigation of splicing and expression across tissues and cell types. The work is well-organized, with informative figure legends and supplemental data that clearly show no substantial phenotypic changes in retinal lamination, proliferation, or cell destiny, despite identified transcriptional and splicing modifications.

Weaknesses:

The retina-specific method raises questions regarding whether Ptbp1 is required in other CNS locations where its neurogenic roles were first proposed. Although the study performs well in transcriptome and histological analyses, it lacks functional assessments (such as electrophysiological or behavioral testing) to determine if small changes in splicing or gene expression affect retinal function.

---

## [Referee Report · Reviewer #2 (Public review)]

Summary:

Ptbp1 has been proposed as a key regulator of neuronal fate through its role in repressing neurogenesis. In this study, the authors conditionally inactivated Ptbp1 in mouse retinal progenitor cells using the Chx10-Cre line. While RNA-seq analysis at E16 revealed some changes in gene expression, there were no significant alterations in retinal cell type composition, and only modest transcriptional changes in the mature retina, as assessed by immunofluorescence and scRNAseq. Based on these findings, the authors conclude that Ptbp1 is not essential for cell fate determination during retinal development.

Strengths:

Despite some effects of Ptbp1 inactivation (initiated around E11.5 with the onset of Chx10-Cre activity) on gene expression and splicing, the data convincingly demonstrate that retinal cell type composition remains largely unaffected. This study is highly significant since it challenges the prevailing view of Ptbp1 as a central repressor of neurogenesis and highlights the need to further investigate, or re-evaluate, its role in other model systems and regions of the CNS.

Weaknesses:

A limitation of the study is the use of the Chx10-Cre driver, which initiates recombination around E11. This timing does not permit assessment of Ptbp1 function during the earliest phases of retinal development, if expressed at that time.

Comments on revisions:

The authors have thoroughly and satisfactorily addressed all my previous comments.

---

## [Author Response]

The following is the authors’ response to the original reviews.

**Reviewer #1 (Public review):**
Summary:The researchers sought to determine whether Ptbp1, an RNA-binding protein formerly thought to be a master regulator of neuronal differentiation, is required for retinal neurogenesis and cell fate specification. They used a conditional knockout mouse line to remove Ptbp1 in retinal progenitors and analyzed the results using bulk RNA-seq, single-cell RNA-seq, immunohistochemistry, and EdU labeling. Their findings show that Ptbp1 deletion has no effect on retinal development, since no defects were found in retinal lamination, progenitor proliferation, or cell type composition. Although bulk RNA-seq indicated changes in RNA splicing and increased expression of late-stage progenitor and photoreceptor genes in the mutants, and single-cell RNA-seq detected relatively minor transcriptional shifts in Müller glia, the overall phenotypic impact was low. As a result, the authors conclude that Ptbp1 is not required for retinal neurogenesis and development, thus contradicting prior statements about its important role as a master regulator of neurogenesis. They argue for a reassessment of this stated role. While the findings are strong in the setting of the retina, the larger implications for other areas of the CNS require more investigation. Furthermore, questions about potential reimbursement from Ptbp2 warrant further research.Strengths:This study calls into doubt the commonly held belief that Ptbp1 is a critical regulator of neurogenesis in the CNS, particularly in retinal development. The adoption of a conditional knockout mouse model provides a reliable way for eliminating Ptbp1 in retinal progenitors while avoiding the off-target effects often reported in RNAi experiments. The combination of bulk RNA-seq, scRNA-seq, and immunohistochemistry enables a thorough examination of molecular and cellular alterations at both embryonic and postnatal stages, which strengthens the study's findings. Furthermore, using publicly available RNA-Seq datasets for comparison improves the investigation of splicing and expression across tissues and cell types. The work is wellorganized, with informative figure legends and supplemental data that clearly show no substantial phenotypic changes in retinal lamination, proliferation, or cell destiny, despite identified transcriptional and splicing modifications.

We thank the Reviewer for their evaluation of the strengths of the study.

Weaknesses:The retina-specific method raises questions regarding whether Ptbp1 is required in other CNS locations where its neurogenic roles were first proposed. The claim that Ptbp1 is "fully dispensable" for retinal development may be toned down, given the transcriptional and splicing modifications identified. The possibility of subtle or transitory impacts, such as ectopic neuron development followed by cell death, is postulated, but not completely investigated. Furthermore, as the authors point out, the compensating potential of increased Ptbp2 warrants additional exploration. Although the study performs well in transcriptome and histological analyses, it lacks functional assessments (such as electrophysiological or behavioral testing) to determine if small changes in splicing or gene expression affect retinal function. While 864 splicing events have been found, the functional significance of these alterations, notably the 7% that are neuronalenriched and the 35% that are rod-specific, has not been thoroughly investigated. The manuscript might be improved by describing how these splicing changes affect retinal development or function.

We have revised the text to address these points as requested.

**Reviewer #2 (Public review):**
Summary:Ptbp1 has been proposed as a key regulator of neuronal fate through its role in repressing neurogenesis. In this study, the authors conditionally inactivated Ptbp1 in mouse retinal progenitor cells using the Chx10-Cre line. While RNA-seq analysis at E16 revealed some changes in gene expression, there were no significant alterations in retinal cell type composition, and only modest transcriptional changes in the mature retina, as assessed by immunofluorescence and scRNAseq. Based on these findings, the authors conclude that Ptbp1 is not essential for cell fate determination during retinal development.Strengths:Despite some effects of Ptbp1 inactivation (initiated around E11.5 with the onset of Chx10-Cre activity) on gene expression and splicing, the data convincingly demonstrate that retinal cell type composition remains largely unaffected. This study is highly significant since it challenges the prevailing view of Ptbp1 as a central repressor of neurogenesis and highlights the need to further investigate, or re-evaluate, its role in other model systems and regions of the CNS.

We thank the Reviewer for their evaluation of the strengths of the study.

Weaknesses:A limitation of the study is the use of the Chx10-Cre driver, which initiates recombination around E11. This timing does not permit assessment of Ptbp1 function during the earliest phases of retinal development, if expressed at that time.

We have revised the text to address the potential limitations of the use of the Chx10-Cre driver in this study.

**Reviewer #1 (Recommendations for the authors):**
(1) The author only selected scRNA-Seq datasets to examine the expression patterns of Ptbp1 in the retina; incorporating immunostaining analysis in the mouse retina is necessary.

Ptbp1 expression patterns in the mouse retina were performed in Fig. 1b-1d, where Ptbp1 expression was analyzed via immunostaining for Ptbp1 protein in Chx10-Cre control and Ptbp1KO retinas at E14, P1, and P30, and are quantified in Fig. 1e.

(2) In Figure 1, Ptbp1 signals were still detected in the KO mice, with the author suggesting that this may indicate cross-reactivity with an unknown epitope. Why is this unknown epitope only detected in the ganglion cell layer? Additional antibodies are needed to confirm the staining results. Furthermore, it is essential to verify the KO at the mRNA level using PCR.

We are unsure of the identity of this cross-reacting epitope, although it might be Ptbp2, which is enriched expressed in immature retinal ganglion cells (Fig. S1). In any case, we do not believe that the identity of this epitope is not relevant to assessing the efficiency of Ptbp1 deletion, as it is not detectably expressed in retinal ganglion cells in any case (Fig. S1).

Although the heatmap in Figure 2B indicates a decrease in Ptbp1 levels in the KO mice, the absence of statistical data makes it difficult to evaluate the KO efficiency.

Respectfully, we believe that Ptbp1 knockout efficiency is adequately addressed using immunohistochemistry, and that further statistical analysis is not essential here.

Cre staining of the Chx10-Cre;Ptbp1lox/lox mice or using reporter lines is also suggested to indicate the theoretically knockout cells. Providing high-power images of the Ptbp1 staining would help readers clearly recognize the staining signals.

To clarify the identity of the knockout cells, we have updated Figure 1 to include the Chx10-CreEGFP staining which more clearly delineates the cells in which Ptbp1 is deleted. Regarding verification of the knockout, we believe additional PCR assays are not necessary, as we have already demonstrated efficient loss of Ptbp1 in Chx10-Cre-expressing cells at the RNA level by both single-cell RNA-sequencing and bulk RNA-sequencing, and also at the protein level by immunohistochemistry. Sun1-GFP Cre reporter lines are also used in Figures 1 and S2 to visualize patterns of Cre activity, a point which is now highlighted in the text. Together, these approaches provide sufficient evidence for effective Ptbp1 knockout.

(3) The possibility of ectopic neuron formation followed by cell death is intriguing but underexplored. Consider adding apoptosis assays (e.g., TUNEL staining) at early developmental stages to test this hypothesis.

While apoptosis assays such as TUNEL staining would be helpful to address this hypothesis, we feel incorporating these additional experiments is currently beyond the scope of this study. We agree the possibility of cell death is intriguing and plan to explore this in future work.

(4) On page 4, the statement "We did not observe any significant differences ... Chx10Cre;Ptbp1lox/lox mice (Fig. 2b,c)" should refer to Fig. 3b,c instead.

We have changed the text to refer to Fig. 3b,c.

(5) The labeling in Figure 3 as "Cre-Ptbp1" is inconsistent with the figure legend "Ptbp1-Ctrl.".

This language was used because the samples for EdU staining in Figure 3 were Chx10-Cre negative Ptbp1^lox/lox^ mice. We have updated the language in the manuscript and figure to reflect the genotypes more clearly.

(6) P30 mice are still sexually immature; the term "adolescent" or "juvenile" should be used instead of "adult."

We have updated the language in the text from “adult” to “adolescent” to describe P30 mice, although the retina itself is mature by this age.

**Reviewer #2 (Recommendations for the authors):**
(1) As mentioned in the public review, a limitation of the study is that Ptbp1 KO is not induced prior to E11. The authors should acknowledge this limitation and include in the Discussion that the use of the Chx10-Cre line does not permit evaluation of a potential role for Ptbp1 during very early stages of retinal development, should it be expressed at that time (an aspect that would be important to determine).

We and have added this limitation to the Discussion in the sentence highlighted below.

Furthermore, the use of the Chx10-Cre transgene in this study does not exclude a potential role for Ptbp1 during very early stages of retinal development prior to E11 (pg. 6).

(2) While the data convincingly show no significant changes in retinal cell type distribution in Ptbp1 mutants, the claims in the abstract and introduction that Ptbp1 is "dispensable for retinal development" or "dispensable for the process of neurogenesis" may be overstated. Indeed, the results indicate that loss of Ptbp1 function influences retinal development by promoting neurogenesis through induction of a neuronal-like splicing program in neural progenitors. Concluding solely that Ptbp1 is dispensable for retinal cell fate specification, rather than for retinal development as a whole, would thus seem more accurate.

We have updated the language in the text to reflect Ptbp1’s role in regulating retinal cell fate specification more clearly.

(3) The authors conclude from Figure 5 that "No changes in the identity or composition of any retinal cell type were observed." Which statistical test was applied to support this conclusion? The figure indicates that Müller cells comprise 10.5% of the total cell population in controls versus 8.2% in Ptbp1-KO retinas. It may be important to consider the overall distribution of glia versus all neurons (rather than each neuron subtype individually). While the observed difference (~2% more glia at the expense of neurons) appears modest, it would be important to determine whether this trend is consistent and statistically significant.

To evaluate cell type composition, we performed differential expression analysis across all major retinal cell types and compared proportional cell type representation between control and Ptbp1 KO retinas. While these analyses did not reveal marked differences in any specific cell type, we acknowledge that the scRNA-Seq dataset includes a single experimental replicate, containing two retinas in each replicate. Therefore, we cannot draw firm statistical conclusions regarding the relative distribution of glia versus neurons, and the modest difference observed in glia cell proportion should be interpreted with caution. We agree that assessing glia-to-neuron ratios across additional replicates will be important in future studies.

(4) Referringx to Figure S1 (scRNA-seq data), the authors state that Ptbp1 mRNA is robustly expressed in retinal progenitors and Müller glia in both mouse and human retina. While the immunostaining in Figure 4 indeed clearly shows strong expression in Müller cells, the scRNAseq data presented in Figure S1 do not support the claim of "robust" expression in Müller glia in the mouse retina. This is even more striking in the human data, where panels F and H show that Ptbp1 is expressed at extremely low, certainly not "robust", levels in Müller cells. The corresponding sentence in the Results section should therefore be revised to more accurately reflect the data presented in Figure S1, or be supported by complementary immunofluorescence evidence.

We thank the reviewer for this comment. We have revised this section of the Results to better reflect Fig S1, as follows:

We observe high expression levels of Ptbp1 mRNA in primary retinal progenitors in both species and Müller glia in mouse retina, with weaker expression in neurogenic progenitors, and little expression detectable in neurons at any developmental age.

(5) When mentioning potential compensation by Ptbp2, the authors may also consider discussing the possibility that compensatory mechanisms can differ between knockdown and knockout approaches. In this context, it is noteworthy that a recent study by Konar et al., Exp Eye Res, 2025 (published after the submission of the present manuscript) reports that Ptbp1 knockdown promotes Müller glia proliferation in zebrafish.

We thank the reviewer for this suggestion. To address this, we have included a section considering this possibility in the discussion section highlighted below.

It is also possible that compensatory mechanisms differ between knockdown and knockout approaches. Notably, a recent study (Konar et al. 2025) reported that Ptbp1 knockdown promotes Müller glia proliferation in zebrafish, suggesting that effects of acute reduction of Ptbp1 may not fully mirror those of complete loss-of-function.

(6) The statistical analyses were performed using a t-test. However, this parametric test is not appropriate for experiments with low sample sizes. A non-parametric test, such as the MannWhitney test, would be more suitable in this context. Furthermore, performing statistical analysis on n = 2 (Figure 3C) is not statistically valid.

We thank the reviewer for this comment. We agree that with a small n, non-parametric tests are more appropriate. We have added additional retinas (now n=5) for the Ptbp1-KO condition in Figure 3C and reanalyzed with the appropriate non-parametric Mann-Whitney test. For all other datasets with sufficient replicates (n≥ 4/genotype), parametric tests such as unpaired t-tests remain valid, and the results are consistent with non-parametric testing.

(7) Figure S3 is accompanied by only a brief explanation in the Results section (a single sentence despite the figure containing six panels), which makes it difficult for readers unfamiliar with this type of data to interpret.

We thank the reviewer for the suggestion. To address this, we have included a more detailed explanation of Supplementary Figure S3 to better clarify our analysis of mature neuronal and glial cell types in both Ptbp1-deficient and wild-type animals. The relevant text now reads:

Notably, splicing patterns in Ptbp1-deficient retinas showed stronger correlation with Thy1positive neurons— which exhibit low Ptbp1 expression—and minimal overlap with microglia and auditory hair cells, the adult cell types with the highest Ptbp1 levels (Fig. S3).

Gene expression and splicing changes were compared across several reference tissues: heart tissue and Thy1-positive neurons, mature hair cells, microglia, and astrocytes (Fig. S3a,b). A heatmap of differentially expressed genes showed that while Ptbp1-deficient retinas diverged from WT retinas, their expression profiles did not resemble those of fully differentiated cell types like rods, astrocytes, or adult WT retina (Fig. S3c). Consistently, Pearson correlation analysis revealed that Ptbp1-deficient and WT retinas were more similar to each other than to fully differentiated neuronal or glial populations (Fig. S3d). Splicing profile analysis further revealed that while there was high correlation of PSI between Ptbp1-deficient and WT retinas, Ptbp1deficient retinas more closely resembled Thy1-positive neurons, whereas WT retinas aligned more strongly with mature cells such as astrocytes, microglia, and auditory hair cells (Fig. S3ef). Together, these results suggest that although Ptbp1 loss induces hundreds of alternative splicing events, the magnitude of PSI changes in the KO retinas remains considerably lower than that seen in fully differentiated cell types (Extended Data 3). Thus, while a subset of splicing events overlaps with those characteristic of mature neurons or rods, the overall splicing and expression profiles of KO retinas are more similar to those of developing retinal tissue rather than terminally differentiated neuronal or glial populations.

(8) To assess progenitor proliferation, the authors performed EdU labeling experiments in P0 retinas. Is there a rationale for not examining earlier developmental time points to evaluate potential effects on early RPCs?

We thank the reviewer for this comment. We chose to perform EdU labeling experiments at P0 for several reasons. P0 represents a developmental stage where RPCs are actively proliferating and represent ~35% of all retina cells, and the retina is transitioning to intermediate-late-stage development, providing sufficient time to ensure efficient and widespread disruption of Ptbp1. Earlier embryonic timepoints were not examined here, as addressing all stages of development was beyond the scope of this current study. However, we agree that investigating whether Ptbp1 plays stage-specific roles during development on early RPCs is an important question and potential future direction.

(9) In Figure S2, panel D shows staining in GCL under the Ptbp1 condition that does not make sense and is inconsistent with panel C. If possible, the authors should provide an alternative image to prevent any confusion.

Thank you for bringing this to our attention. The image shown for Ptbp1-KO in Figure 2d shows Sun1-eGFP labeling, which labels every cell affected by the Cre condition. The genotype for this mouse was Chx10-Cre;Ptbp1lox/lox;Sun1-GFP. We apologize for any confusion and have updated the genotype in the figure legend.

(10) The authors should revise the following sentence at the end of the Discussion section, as its meaning is unclear: "...and conditions for in vitro analysis may have accurately replicated conditions in the native CNS."

We thank the reviewer for this comment and have revised this sentence in the discussion for the sentence below.

Previous studies using knockdown may have been complicated by off-target effects (Jackson et al. 2003), and conditions for in vitro analysis may not have accurately replicated conditions in the native CNS.